# A Novel Preclinical In Vitro 3D Model of Oral Carcinogenesis for Biomarker Discovery and Drug Testing

**DOI:** 10.3390/ijms24044096

**Published:** 2023-02-17

**Authors:** Ravi Teja Chitturi Suryaprakash, Kate Shearston, Camile S. Farah, Simon A. Fox, Muhammad Munir Iqbal, Ulrich Kadolsky, Xiao Zhong, Alka Saxena, Omar Kujan

**Affiliations:** 1UWA Dental School, The University of Western Australia, Nedlands, WA 6009, Australia; 2Australian Centre for Oral Oncology Research and Education, Nedlands, WA 6009, Australia; 3Harry Perkins Institute of Medical Research, QEII Medical Centre and Centre for Medical Research, The University of Western Australia, Nedlands, WA 6009, Australia; 4Genomics WA, Harry Perkins Institute of Medical Research, Telethon Kids Institute, The University of Western Australia, Nedlands, WA 6009, Australia

**Keywords:** oral squamous cell carcinoma, oral epithelial dysplasia, spheroid, chemoprevention, tobacco, alcohol, VEGF, pazopanib, lenvatinib

## Abstract

This study aimed to develop an in vitro three-dimensional (3D) cell culture model of oral carcinogenesis for the rapid, scalable testing of chemotherapeutic agents. Spheroids of normal (HOK) and dysplastic (DOK) human oral keratinocytes were cultured and treated with 4-nitroquinoline-1-oxide (4NQO). A 3D invasion assay using Matrigel was performed to validate the model. RNA was extracted and subjected to transcriptomic analysis to validate the model and assess carcinogen-induced changes. The VEGF inhibitors pazopanib and lenvatinib were tested in the model and were validated by a 3D invasion assay, which demonstrated that changes induced by the carcinogen in spheroids were consistent with a malignant phenotype. Further validation was obtained by bioinformatic analyses, which showed the enrichment of pathways associated with hallmarks of cancer and VEGF signalling. Overexpression of common genes associated with tobacco-induced oral squamous cell carcinoma (OSCC), such as MMP1, MMP3, MMP9, YAP1, CYP1A1, and CYP1B1, was also observed. Pazopanib and lenvatinib inhibited the invasion of transformed spheroids. In summary, we successfully established a 3D spheroid model of oral carcinogenesis for biomarker discovery and drug testing. This model is a validated preclinical model for OSCC development and would be suitable for testing a range of chemotherapeutic agents.

## 1. Introduction

Oral squamous cell carcinoma (OSCC) is a malignant neoplasm that arises from the epithelial cells lining the mucosa of the oral cavity [1]. OSCC is a global health burden affecting more than 377 thousand people around the globe [2]. In Australia, OSCC incidence is projected to increase by 50% in 2030 [3]. Several risk factors are associated with the development of OSCC, but the most common and significant are tobacco use and alcohol consumption [4]. OSCC frequently arises from oral potentially malignant disorders (OPMDs); however, there are very few studies that have studied the effect of chemotherapeutic agents in preventing OPMDs from undergoing malignant transformation, leaving a substantial research gap [5,6]. There has been considerable research on therapeutics for head and neck cancer, with drugs such as cetuximab being approved for use [7]. The identification of druggable targets for OSCC is also a research focus [8], but there is limited research taking place to identify targets that prevent OPMDs from undergoing transformation. In vitro cell culture preclinical models are a simple and effective method to investigate these targets.

To date, traditional cell culture experiments have been used for this purpose, with cells cultured on flat surfaces [9], but such methods do not accurately represent the tumor microenvironment (TME). The TME is more effectively created using three-dimensional (3D) cell culture methods, the simplest of which are spheroids [10,11]. Animal models of oral carcinogenesis, for example, in the mouse [12] or hamster [13], can also be used but can be challenging to translate to human clinical studies and are associated with handling and ethical challenges [14]. There is also a need to replace, reduce and refine the use of animal models for research purposes [15].

Next-generation sequencing has improved the understanding of various aspects of cell biology, including important components of normal and disease processes [16]. Transcriptomics specifically has helped to understand the process of carcinogenesis in many human cancers, including lung, thyroid, and colon [17,18,19]. As there are very limited studies that have used this powerful technology to study oral carcinogenesis, we aimed to combine both these influential tools, i.e., 3D cell culture technology and transcriptomic analysis, to identify potential pathways involved in the progression of OSCC. We used the well-known carcinogen, 4-Nitroquinoline N-oxide (4NQO), which replicates the synergistic effects of tobacco and alcohol [20]. As a proof of concept, we also assessed the utility of the spheroid carcinogenesis model as a method for testing the effectiveness of inhibitors of potential druggable targets.

## 2. Results

**Morphometric (all measurements in µm) and viability analysis (all units are absorbance values at 405 nm):** Over the course of the treatment with 4NQO, HOK and DOK spheroids remained fairly consistent in size. The average size of HOK spheroids was 663.9 ± 32.9 µm on day 1, and this was reduced to a size of 389.7 ± 26.7 µm and 375.1 ± 4.6 µm for the low and high doses of 4NQO, respectively. As expected, the viability of the treated HOK spheroids reduced from 1.142 ± 0.16 (low dose) and 1.346 ± 0.11 (high dose) on day 1 to 0.38 ± 0.10 (low dose) and 0.37 ± 0.55 (high dose) on day 8 compared to controls (1.25 ± 0.21 on day 1 to 1.05 ± 0.05 on day 8) (Figure 1). Similarly, DOK spheroids were 587.2 ± 45.1 µm on day 1 and reduced to 536.5 ± 35.7 µm for the low dose of 4NQO and 548.75 ± 44.32 µm for the high dose of 4NQO. The viability of the treated spheroids also reduced at the conclusion of treatment with the carcinogen 0.50 ± 0.17 (low dose) and 0.37 ± 0.13 (high dose) when compared to the controls (0.68 ± 0.10) (Figure 2).

**3D invasion assay:** At the conclusion of treatment with the carcinogen, both HOK and DOK spheroids treated with the high dose of 4NQO showed consistent and increased properties of invasion as evidenced by the length as well as the area of invasion when compared to the controls and the low-dose-treated spheroids. The average length of invasion was significantly increased (*p* = 0.002) in both high-dose-treated HOK spheroids (296.7 ± 62.1 µm) and low-dose-treated HOK spheroids (285.8 ± 30.1 µm) compared to controls (247.0 ± 32.1 µm). The average area of invasion was significantly increased (*p* = 0.001) in high-dose-treated HOK spheroids (52,050.5 ± 1031.2 µm^2^) compared to the low-dose-treated HOK spheroids (48,452.4 ± 1063.7 µm^2^) and controls (46,705.1 ± 1389.4 µm^2^) as shown in Table 1 and Figure 1. Similarly, in experiments with DOKs, the average length of invasion in high-dose DOK spheroids (563.3 ± 42.5 µm) was significantly increased (*p* = 0.001) compared with low-dose-treated DOK spheroids (429.3 ± 54.4 µm) and controls (361.5 ± 87.6 µm). The area of invasion in both high-dose-treated DOK spheroids (119,331.06 ± 14,618.64 µm^2^) and low-dose-treated DOK spheroids (101,822.2 ± 1842.1 µm^2^) was significantly higher (*p* = 0.001) compared to controls (62,749.8 ± 5463.1 µm^2^) (Figure 2 and Table 1).

**RNASeq analysis:** RNASeq analysis showed that when compared to the controls, the high-dose 4NQO-treated HOK spheroids exhibited 157 up-regulated genes and 354 down-regulated genes (Appendix A). In the DOK spheroids, there were a total of 126 upregulated and 130 downregulated genes in the high-dose-treated spheroids compared to controls (Appendix A). Hierarchical clustering of the top upregulated and downregulated genes in HOK and DOK spheroids is shown in Appendix A, Figure 1 and Figure 2. Selected upregulated pathways in treated HOK and DOK spheroids are listed in Table 2. The most significantly upregulated pathway in treated HOK spheroids was the Roundabout4—vascular endothelial growth factor (Robo4-VEGF signalling) pathway (*p* = 0.004). Other upregulated pathways included the Notch, epithelial-to-mesenchymal transition, rat sarcoma (RAS), focal adhesion, phosphoinositide 3-kinase/Ak strain-transforming (PI3K/AKT), and the xenobiotic metabolism pathways. In treated DOK spheroids, the apoptotic pathway was downregulated, and the following pathways were upregulated: KRAS, neovascularisation processes, matrix metalloproteinases, Robo4-VEGF signaling pathways crosstalk, benzo(a)pyrene metabolism, and PI3K/AKT/mammalian target of rapamycin (mTOR) signaling pathways. The complete list of upregulated and downregulated pathways for both HOKs and DOKs is presented in Appendix A.

**Expression of VEGF pathway in oral carcinogenesis**: To provide additional supporting evidence, we examined selected genes of the Robo4-VEGF signaling pathway. Quantitative assessment of VE-cad, VEGF, VEGFR1 and VEGFR2 by qPCR showed that expression of these genes increased with the treatment of cells with 4NQO. This was further confirmed at the protein level for VEGF and YAP1 by immunohistochemistry using an independent cohort of OSCC, dysplasia and epithelial hyperkeratosis. A total of 38 FFPE blocks were included for validation purposes. Overall, 9 (23.7%) cases were of epithelial hyperkeratosis, 11 (28.9%) cases were diagnosed with low-risk OED, 10 (26.3%) cases were diagnosed as high-risk OED, and 8 (21.1%) cases were of OSCC. Positive staining of cytoplasmic expression was identified in all cases (Figure 3). The IRS score of VEGF was independently significantly associated with the grade of epithelial disorders (*p* = 0.0002) (Figure 4). A Spearman’s rank-order correlation showed that the direction of this association was strongly positive from epithelial hyperkeratosis to OSCC, rs = 0.773. Post hoc analyses revealed that the IRS significantly increased among the following pairwise: (i) epithelial hyperkeratosis and low-risk OED (*p* = 0.0002), (ii) epithelial hyperkeratosis and high-risk OED (*p* < 0.001), (iii) epithelial hyperkeratosis and OSCC (*p* = 0.0003), and (iv) low-risk OED and OSCC (*p* = 0.002) (Figure 4). Moreover, all cases stained against YAP1 showed positive immunoreactivity of cytoplasmic and/or membranous expression (Figure 3). The score of expression was associated with the grade of epithelial disorders in a statistically significant pattern, as shown in Figure 4 (*p* = 0.0046). The direction of this association was strong and positively increased from epithelial hyperkeratosis to OSCC, rs = 0.898. Tukey’s post hoc analyses showed statistically significant differences in all pairwise comparisons (*p* = 0.0004) except between hyperkeratosis and low-risk OED (*p* = 0.636), as shown in Figure 4.

**Chemoprevention assay:** The 3D invasion assay embedded HOK and DOK transformed (high dose 4NQO treated) spheroids in Matrigel and simultaneously treated them with the inhibitors pazopanib and lenvatinib to assess the chemopreventive effects of these drugs. Both the length and area of invasion were significantly decreased (*p* = 0.01) in the transformed spheroids treated with pazopanib or lenvatinib when compared to untreated transformed spheroids, which acted as controls (Table 3). In the HOK experiments, the invasion length was reduced by 55.9% for pazopanib (153.8 ± 29.3 µm) and 62.4% for lenvatinib (131.1 ± 20.4 µm)-treated spheroids compared to controls (341.6 ± 40.9 µm). The invasion area was also reduced by 61.7% for pazopanib (14,862.8 ± 2683.2 µm^2^) and 69.5% for lenvatinib (15,486.0 ± 4136.3 µm^2^)-treated spheroids compared to controls (50,718.3 ± 814.0 µm^2^) (Figure 5 and Table 3). Similarly, the pazopanib-treated DOK spheroids showed a mean invasive length of 373.8 ± 49.3 µm, which was a reduction of 39.2%, and the lenvatinib-treated spheroids showed a mean invasive length of 341.3 ± 20.0 µm, which was a reduction of 44.5% compared to controls (614.2 ± 39.4 µm). The invasion area was also reduced by 43.8% with pazopanib (71,685.5 ± 16,675.1 µm^2^) and by 61.2% in lenvatinib (49,848.6 ± 27,842.6 µm^2^)-treated spheroids compared to controls (125,235.7 ± 3158.1 µm^2^) (Figure 5 and Table 3).

## 3. Discussion

Oral carcinogenesis is a complex process. Identifying pathways for the progression and testing of chemopreventive agents is an important and underexplored aspect of oral carcinogenesis research, and in vitro studies are promising tools for this purpose. In this study, we aimed to develop an in vitro 3D cell culture model of oral carcinogenesis for the rapid, scalable testing of chemotherapeutic agents. In this process, our first objective was to establish and validate the model of oral carcinogenesis involving the transformation of normal and dysplastic epithelial cells to cancer by treating them with a well-known carcinogen, 4NQO. HOK and DOK cells were treated with low and high doses of 4NQO in their spheroid culture. To validate our model, we assessed malignant changes both phenotypically and genotypically. The phenotypic assessment of malignant transformation was carried out using a 3D invasion assay at the conclusion of treatment. As expected, the treated spheroids showed phenotypic changes correlating with malignant transformation, evidenced by significantly increased invasion (in both length and area) into the Matrigel when compared to controls. The spheroids treated with the high dose of 4NQO showed more consistent and significantly increased invasive properties when compared with the low dose of 4NQO. Thus, we selected RNA extracted from high-dose 4NQO-treated spheroids for bioinformatic analyses. These transformational phenotypic changes were reflected in the RNASeq data with the enrichment of pathways related to epithelial-mesenchymal transition (EMT) and focal-adhesion molecules. These observations were consistent with a recent review of the molecular landscape of OSCC [7]. The increased invasive capabilities of malignant cells can also be attributed to the overexpression of matrix metalloproteinases, such as MMPs 1, 3 and 9, which are all well-established biomarkers elevated in OSCC, further validating our model [21,22,23,24]. Furthermore, upregulated xenobiotic and benzo(a)pyrene metabolism pathways and the overexpression of CYP1A1 and CYP1B1 indicate that our model accurately represents tobacco and alcohol-associated OSCC [25]. Cancer cells are characterised by an increase in glucose metabolism when compared to normal cells and a metabolic shift from oxidative phosphorylation to glycolysis [26]. The pyruvate dehydrogenase complex and pyruvate dehydrogenase kinase (PDK) isoforms 1–4 are considered to be the main regulators of this metabolic shift, and in our study, we found that *PDK4* was one of the genes that were overexpressed. *PDK4* has been found to be over-expressed and implicated in the oncogenesis of many cancers, including that of the colon and bladder [27,28,29]. The role of PDKs is understudied in OSCC and could potentially be an important target that could be assessed as a biomarker for malignant transformation [30,31]. Resisting cell death via apoptosis is also a well-known hallmark of cancer, and the role of caspases in this regard is well established [32,33]. In our study, bio-informatic analyses showed that apoptosis pathways and genes such as CASP10, which encodes caspase-10, were significantly downregulated, as seen in malignant tumors, adding further evidence for the validity of our model [34]. The Notch pathway was also significantly upregulated when normal spheroids were treated with the carcinogen. This probably indicates that this pathway may be involved in the early stages of oral carcinogenesis. The involvement of Notch signalling has been implicated in differentiating between various squamous cell carcinomas [35]. Some of the genes upregulated in this pathway include *DLK1* and *DLX2*. These genes have been found to be involved in the malignant transformation and progression of many tumours, including OSCC [36,37,38]. All these results provide compelling evidence that our preclinical model is valid and accurately represents tobacco-induced OSCC.

In line with the aim of our study, the next objective was to identify some of the potential pathways that could be involved in the progression to OSCC. As in many cases of OSCC, there appears to be a step-wise progression from normal to dysplasia and cancer. Therefore, we examined some of the common pathways that were enriched between the treatment of both normal and dysplastic spheroids, with the carcinogen correlating with both early and late events occurring in progression. We found that the RAS and PI3K-Akt pathways were commonly enriched in both treatments, which is consistent with the literature [39,40,41]. Although these pathways were upregulated in both groups, they did not reach significance. Interestingly, in both groups, many of the pathways associated with tumor angiogenesis (VEGFA-VEGFR2, Robo4 and VEGF crosstalk, angiogenesis, and neovascularisition) were significantly upregulated. It appears that VEGF signalling via VEGFR plays a significant role in the 4NQO-mediated progression of OSCC. This finding is consistent with data already present in the literature [42]. The VEGF family consists of 7 members, including VEGF and VEGFR, and plays a significant role in tumor angiogenesis [43]. Interestingly, our results showed an increase in the mRNA expression of selected genes of the VEGF family after treating the HOK and DOK cells with 4NQO. This observation supports the role of the VEGF pathway in oral carcinogenesis. In our study, changes in VEGF expression promoted cell invasion/metastasis by activating the transcription of matrix metalloproteinase 1 (MMP1), MMP2, MMP9, and MMP10 [44]. Likewise, the mRNA expression of VE-cadherin increased significantly in the treated HOK and DOK cells following a similar increase in the VEGF mRNA expression. In previous studies, the association between VE-cadherin and VEGF was found to be critical in tumorigenesis because VE-cadherin has been shown to interact with β-catenin, forming a complex with VEGFR-2 and PI3-kinase that is required to trigger antiapoptotic pathway [45]. Furthermore, the VEGF pathway interacts with the EMT pathway, contributing to OSCC development [7]. YAP1 is considered an oncogene associated with EMT, tumor cell proliferation, and metastasis [46]. It is believed that VEGF works through YAP1 activation [47]. Hence, we further validated the changes in the expression of VEGF and YAP1 in different cohorts of lesions representing the several stages in OSCC development using immunohistochemistry. Our findings support that angiogenic changes are associated with the severity of the epithelial disorder. It is more likely that VEGF acts as a critical mediator of tumor angiogenesis by stimulating the growth of new microvessels supporting other pathways, such as EMT and hypoxia, to sustain oncogenesis [43,48].

A final objective was to test if our model was clinically relevant to test drugs as chemopreventive agents. An ideal model of carcinogenesis should be able to test chemotherapeutic agents. For this purpose, two inhibitors associated with VEGF-VEGFR signalling, pazopanib and lenvatinib, were selected. Targeting angiogenesis has become an attractive therapeutic strategy for several types of cancer, including head and neck SCC [49,50]. A recent study demonstrated supporting evidence of the efficacy of apatinib mesylate, a novel anti-angiogenic agent, in the management of recurrent/metastatic inoperable HNSCC [50]. We tested the efficacy of pazopanib and lenvatinib on transformed spheroids [51,52]. The doses selected corresponded to the maximum serum concentration that the drug would achieve in the body after administration (C_max_). The doses for both these drugs were 22.4 µg/mL and 20 nM for pazopanib and lenvatinib, respectively. These doses were selected based on data from the literature and the Australian Therapeutic Goods Administration [53,54]. Both these drugs inhibited the invasion of transformed spheroids as expected, presumably by blocking VEGFR. Thus, our model appears to not only accurately represent tobacco-induced OSCC but can also be used as a preclinical tool to test chemopreventive agents before starting clinical trials. Some of the other advantages of our model are that it can be used as a rapid tool to mimic oral carcinogenesis compared to animal models, which require a longer period of time [20]. Specific genes can also be inactivated by gene silencing edited with Clustered Regularly Interspaced Short Palindromic Repeats (CRISPR) to test the role of specific genes in oral carcinogenesis [55]. The chemoprevention assay can be easily scaled to facilitate the automated screening of many drugs. One of the limitations of our study was that spheroids could not be cultured for a longer duration of time, which might be much more useful to replicate OSCC in the lab.

In conclusion, we successfully established a 3D preclinical model of oral carcinogenesis by using normal and dysplastic spheroids and treating them with the known carcinogen 4NQO. We found that pazopanib and lenvatinib have chemopreventative properties in our model since they target the VEGF pathway. Our study supports the clinical use of spheroids in evaluating drugs for oral cancer chemoprevention. With the help of these models, the process of selecting the appropriate drug choice for clinical trials becomes evidence-based.

## 4. Methods and Materials

### 4.1. Cell Culture and Reagents

All reagents were purchased from Sigma-Aldrich, St. Louis, MO, USA, unless otherwise mentioned. Human oral keratinocytes (HOK) were purchased from ScienceCell Research Laboratories and cultured in Dulbecco’s modified Eagle’s medium: Nutrient Mixture F-12 (DMEM: F12) (Invitrogen, Waltham, CA, USA) supplemented with 10% fetal bovine serum (FBS), 400 ng/mL hydrocortisone and 1% antibiotic-antimycotic (ABAM). Human dysplastic oral keratinocytes (DOK) were supplied by the European Collection of Authenticated Cell Cultures (ECACC), grown in Gibco Advanced Dulbecco’s modified Eagle’s medium (ADMEM) (Invitrogen, CA, USA) supplemented with 2% Gibco GlutaMAX (Invitrogen, Waltham, CA, USA), 3% FBS, 10.3 μM hydrocortisone and 1% ABAM. A concentration of 0.36 μM (low dose) and 0.72 μM (high dose) 4NQO was used as the test agent. A total of 22.4 µg/mL pazopanib and 20 nM lenvatinib were used as test doses for chemoprevention assays. Acid phosphatase (APH) buffer was prepared as described previously [56].

### 4.2. Spheroid Culture and Treatment with 4NQO

To generate spheroids, a seeding density of 20,000 cells/well was used for HOKs and 10,000 cells/well for DOKs in 96-well ultra-low attachment (ULA) plates (Corning, New York, NY, USA). HOK and DOK spheroids were exposed to low and high doses of 4NQO for 90 min every day from day 1 to day 8 for HOKs and day 2 to day 6 for DOKs. All experiments were undertaken in triplicate.

### 4.3. Spheroid Imaging and Morphometric Analysis

Spheroids were imaged under phase contrast on a Nikon 7200SR microscope (Nikon, Melville, NY, USA). Images were captured using a high-resolution digital camera, and the spheroid diameter was calculated using the Nikon Elements software (Version 5.21.00, Nikon, Melville, NY, USA). Imaging and morphometric analysis were undertaken during the period when the cells maintained their viability with no signs of necrosis over days 1–8 for all experiments involving HOKs and days 2–6 for all experiments involving DOKs.

### 4.4. Spheroid Viability

The acid phosphatase (APH) assay was used to determine the viability of HOK and DOK spheroids [56]. Briefly, once the treatment with 4NQO was complete, the culture medium was carefully removed, followed by rinsing of the spheroids with phosphate-buffered saline (PBS). APH buffer was added, after which the spheroids were incubated for 4 h. At the end of incubation, 15 μL of 1 M NaOH was added, and absorbance was read at 405 nm.

### 4.5. 3D Invasion Assay

Invasion assays were performed as described elsewhere [57] using Matrigel (BD Biosciences, Franklin Lakes, NJ, USA) containing all growth factors. Briefly, 50 μL of Matrigel was added, making sure spheroids were in the center of the ULA plate. Once the Matrigel was added, plates were incubated for 60 min to allow the Matrigel to polymerize. Following this, culture medium was added, and the spheroids were imaged daily from day 0 to day 4. For chemoprevention assays, the appropriate concentration of inhibitors was added to high-dose carcinogen-treated spheroids in Matrigel. The invasion was quantified at the end of day 4 by (i) measuring the length of invasion from the center of the spheroid and (ii) measuring the area of invaded cells (total area of invaded spheroid − the actual area of the spheroid).

### 4.6. RNA Extraction, RNASeq and Bioinformatics

Following treatment, at least 8 spheroids per condition were harvested for RNA extraction using the Bioline RNA Micro Kit (London, UK) following the manufacturer’s instructions. Briefly, spheroids were harvested along with the culture medium in a microcentrifuge tube, after which the medium was carefully removed and washed with ice-cold PBS. RNA quantity and quality were assessed using a NanoDrop spectrophotometer (ThermoFisher, Waltham, MA, USA), Qubit fluorometer (ThermoFisher, Waltham, MA, USA) and 2100 Bioanalyzer (Agilent Technologies, Santa Clara, CA, USA). RNA from 0.72 µM 4NQO-treated HOK and DOK spheroids and their corresponding controls were selected for RNASeq. The libraries were prepared using the SureSelect XT HS2 RNA System (Agilent Technologies, Santa Clara, CA, USA). The quality of libraries was checked using shallow sequencing on the iSeq 100 System (Illumina, San Diego, CA, USA) before deep sequencing on NovaSeq 6000 (Illumina, CA, USA). Reads were demultiplexed using bcl2fastq v2.20.0.422 (Illumina, San Diego, CA, USA). Low-quality reads were trimmed before alignment using Burrows Wheeler Alignment tool (BWA V0.7.17) vendor-recommended practices and mapped to human reference (hg38) [58]. Aligned reads were de-duplicated using LocatIt (Agilent AGeNT v2.0.5), and differentially expressed genes were called with Partek Flow software v10.0.21.0411 using the R/Bioconductor package DESeq2 with a negative binomial model [59]. Analysis was performed using standard parameters with the independent filtering function enabled to filter genes with low mean normalized counts. Adjusted *p*-values (Padj) for multiple testing, using Benjamini–Hochberg tests to estimate the false discovery rate (FDR), were calculated for the final estimation of DE significance using a fold change of <−2 or >2.

### 4.7. Gene Set Enrichment Analysis for Biological Pathways

Over-representation analysis (ORA) of differentially expressed genes for pathways and gene ontology terms was performed using the web-based platform EnrichR (https://maayanlab.cloud/Enrichr/, accessed on 12 November 2022), which permits the interrogation of multiple databases [60]. Upregulated and downregulated gene lists were evaluated for significant enrichment against the following gene set libraries: the WikiPathways 2019 Human Database [61], the Molecular Signature Database (MSigDB) [62], and KEGG 2019 pathways [63]. Enriched annotations/pathways were selected and ranked based on the combined score, which was calculated by the EnrichR platform following Z-score permutation background correction on the Fisher exact test’s *p*-value [60].

### 4.8. Reverse Transcription Quantitative Real-Time Polymerase Chain Reaction (RT-qPCR)

Total RNA was extracted from 0.72 µM 4NQO-treated HOK and DOK spheroids and their corresponding controls. RNA extraction and quality control were performed as we described above. Gene-specific polymerase chain reaction (PCR) primers for vascular endothelial cadherin (VE-cad), vascular endothelial growth factors (VEGF), vascular endothelial growth factor receptor 1 (VEGFR1), and vascular endothelial growth factor receptor 2 (VEGFR2) were designed using the Primer 3 software (Appendix A). Reverse transcription and qRT-PCR was performed using the SsoAdvanced SYBR Green SuperMix (Bio-Rad, Hercules, CA, USA) according to the manufacturer’s instructions. Data analysis was performed using the CFX Manager software 3.1 (Bio-Rad). GAPD was used as a reference gene to normalise the mRNA expression of the selected genes. Results are presented as deltaCq (ΔCq), calculated by subtracting the Cq value for the target gene from the mean Cq value of the reference genes. Statistical analysis of qPCR data was performed using Prism 6.07 (Graphpad Software, San Diego, CA, USA).

### 4.9. Histopathology and Immunoreactivity Assessment

For validation, the relationship between the immunoreactivity score index (IRS) and the presence of VEGF and yes-associated protein 1 (YAP1) was examined. Archival formalin-fixed paraffin-embedded (FFPE) oral tissue samples of patients’ biopsies at the UWA Dental School were used. The archival materials are unidentified and without demographic details. Ethics approval was obtained from the Human Ethics Committee at the University of Western Australia (RA/4/1/8562). Two sections (4 µm each) were cut from each FFPE, one for H&E staining and the other for immunohistochemical staining to assess the expression of VEGF and YAP1. All H&E slides were assessed to confirm the microscopic diagnosis of all cases. The binary system of grading cases with oral epithelial dysplasia (OED) was used where applicable [64]. Accordingly, the cases were grouped into four categories according to their grade of epithelial disorder (1) epithelial hyperkeratosis, (2) low-risk OED, (3) high-risk OED, and (4) OSCC. Standard immunohistochemistry (IHC) using an IHC detection kit (Pierce Peroxidase IHC, Cat # 3600, ThermoFisher, Waltham, MA, USA) was performed to detect an antibody against VEGF (Cat # MA1-16629, diluted 1:200, Invitrogen, Waltham, MA, USA) and an antibody against YAP1 (Cat # ab52771, diluted 1:200, Abcam, Cambridge, UK). The expression of the antibodies of interest was assessed for each case to obtain an individual IRS as described previously [65]. All cases were assessed blindly by two independent assessors based on ten random fields of at least 500 cells each at a final magnification ×400. Any discrepancy of more than 5% among the assessors was resolved by discussion.

### 4.10. Statistical Analysis

Statistical analysis was performed with GraphPad Prism version 6.07 (GraphPad Software, San Diego, CA, USA) and IBM SPSS Statistics (version 28, IBM Corp., Armonk, NY, USA). The comparisons of groups for 3D invasion assay, morphometric and viability analyses (control vs. low-dose 4NQO-treated and control vs. high-dose 4NQO-treated) and chemoprevention assay (control vs. pazopanib-treated and control vs. lenvatinib-treated) were carried out using the one-way ANOVA test with a *p*-value set to <0.05 for significance. For the evaluation of VEGF and YAP1 immunohistochemistry, the mean and standard deviation (SD) of IRS data for each diagnostic group were compared with other groups using one-way ANOVA. Further post hoc analysis was performed for pairwise comparisons. The strength and direction between the IRS and the lesion severity were established by Spearman’s product-moment correlation coefficient (*r_s_*). For this purpose, the different grades of epithelial disorders were ordinarily graded as (i) epithelial hyperkeratosis = 1, low-risk OED = 2, high-risk OED = 3, and OSCC = 4. Spearman’s coefficient (*r_s_*) was interpreted as weak = 0.1 to 0.39, moderate = 0.4 to 0.69, strong = 0.7 to 0.89, and very strong = 0.9 to 1 [66]. The level of significance was set as *p* < 0.05.

## Figures and Tables

**Figure 1 ijms-24-04096-f001:**
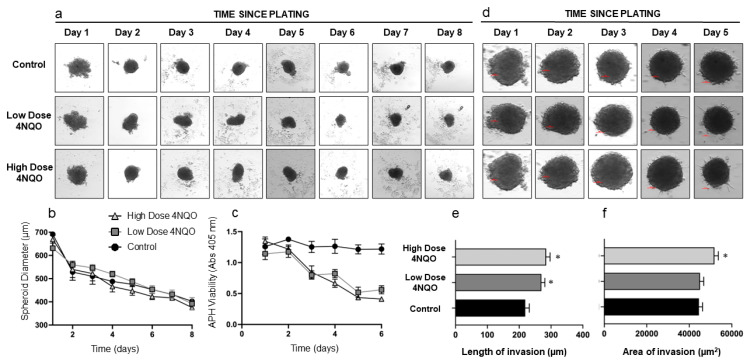
Size, viability, and invasiveness of HOK spheroids treated with the carcinogen 4NQO. (**a**) Representative images of HOK spheroids on days 1–8, treated daily for 90 min with low (0.36 μM) or high (0.72 μM) doses of 4NQO. Cells were plated in low-adherence plates at 20,000 cells per well, and media was changed daily. (**b**) Diameter of control and 4NQO-treated spheroids measured in μm. (**c**) Viability of control and 4NQO-treated spheroids assessed using APH assay (absorbance read at 405 nm). (**d**) Representative images of a Matrigel invasion assay using control and 4NQO-treated spheroids on days 0–4. (**e**) Invasiveness of 4NQO-treated spheroids as measured by the length of invasion (μm). (**f**) Invasiveness of 4NQO-treated spheroids as measured by area of invasion in μm^2^. All error bars are mean ± SEM, and significance is assessed as * *p* < 0.05.

**Figure 2 ijms-24-04096-f002:**
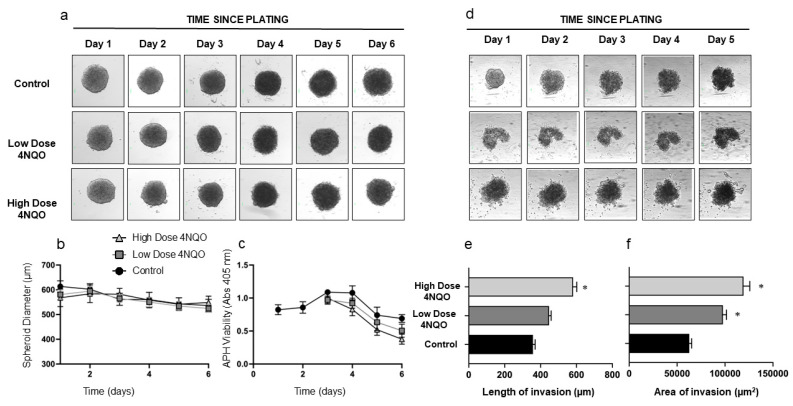
Size, viability, and invasiveness of DOK spheroids treated with the carcinogen 4NQO. (**a**) Representative images of DOK spheroids on days 1–6, treated daily for 90 min with low (0.36 μM) or high (0.72 μM) doses of 4NQO. Cells were plated in low-adherence plates at 10,000 cells per well, and media was changed daily. (**b**) Diameter of control and 4NQO-treated spheroids measured in μm. (**c**) Viability of control and 4NQO-treated spheroids assessed using APH assay (absorbance read at 405 nm). (**d**) Representative images of a Matrigel invasion assay using control and 4NQO-treated spheroids on days 0–4. (**e**) Invasiveness of 4NQO-treated spheroids as measured by length of invasion (μm). (**f**) Invasiveness of 4NQO-treated spheroids as measured by area of invasion in μm^2^. All error bars are mean ± SEM, and significance is assessed as * *p* < 0.05.

**Figure 3 ijms-24-04096-f003:**
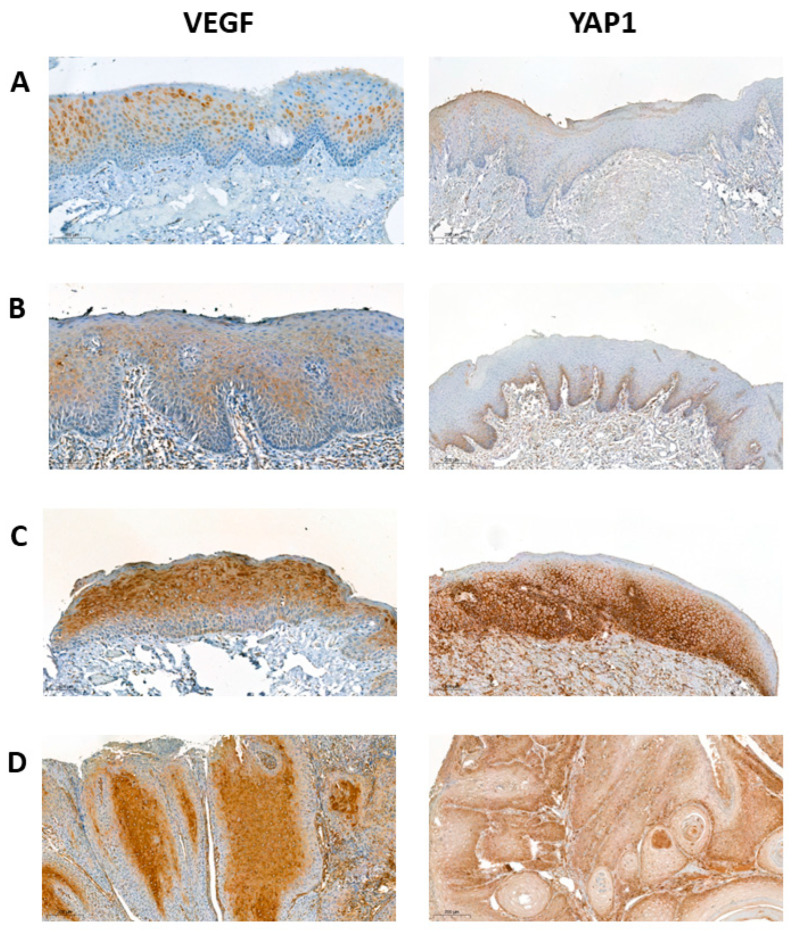
Representative photomicrographs show VEGF and YAP1 expression among (**A**): epithelial hyperkeratosis, (**B**): low-risk OED, (**C**): high-risk OED, and (**D**): OSCC (magnification ×100).

**Figure 4 ijms-24-04096-f004:**
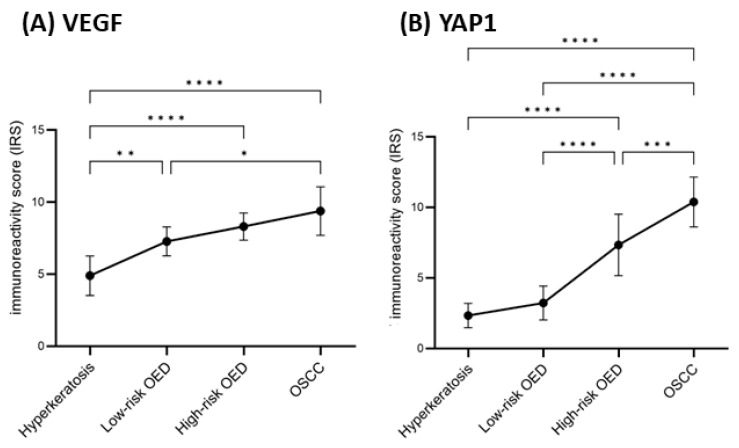
The immunoreactivity score (IRS) of (**A**) VEGF and (**B**) YAP1 among different grades of epithelial disorders and the level of significance for pairwise comparisons. * *p* < 0.05, ** *p* < 0.005, *** *p* < 0.001, **** *p* < 0.0001. The IRS data are reported as mean and standard deviation.

**Figure 5 ijms-24-04096-f005:**
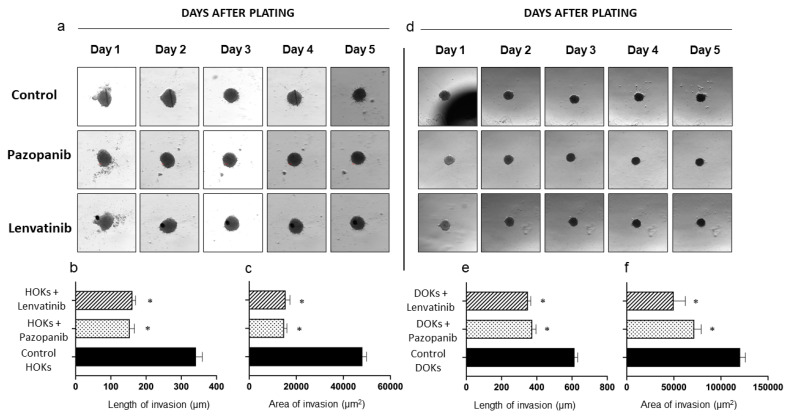
Chemoprevention assay on transformed (0.72 µm 4NQO-treated) HOK and DOK spheroids treated with the inhibitors pazopanib and lenvatinib. (**a**) Representative images of Matrigel chemoprevention assay using inhibitors pazopanib and lenvatinib compared to controls (transformed HOK spheroids) on days 0–4. (**b**) Invasiveness of treated and untreated HOK spheroids as measured by the length of invasion (μm). (**c**) Invasiveness of treated and untreated HOK spheroids as measured by area of invasion in μm^2^. (**d**) Representative images of Matrigel chemoprevention assay using inhibitors pazopanib and lenvatinib compared to controls (transformed DOK spheroids) on days 0–4. (**e**) Invasiveness of treated and untreated DOK spheroids as measured by length of invasion (μm). (**f**) Invasiveness of treated and untreated DOK spheroids as measured by area of invasion in μm^2^. All error bars are mean ± SEM, and significance is assessed as * *p* < 0.05.

**Table 1 ijms-24-04096-t001:** Quantitative assessment of invasion of HOK and DOK spheroids treated with low and high doses of 4NQO using one-way ANOVA.

	HOK Spheroids		DOK Spheroids	
	Length (µm)	Area (µm^2^)	Length (µm)	Area (µm^2^)
**Control**	247.0 ± 32.1	46,705.18 ± 1389.4	361.5 ± 87.6	62,749.8 ± 5463.1
**Low-dose-treated**	285.8 ± 30.1 *	48,452.4 ± 1063.7	429.3 ± 54.4	101,822.2 ± 1842.1 *
**High-dose-treated**	296.7 ± 62.1 *	52,050.5 ± 1031.2 *	563.3 ± 42.5 *	119,331.0 ± 14,618.3 *

* *p*-value less than 0.05.

**Table 2 ijms-24-04096-t002:** Selected pathways enriched by treatment of HOK and DOK spheroids treated with high-dose 4NQO.

	S No.	Pathways	Regulation	*p*-Value
**HOK Spheroids**	1	Robo4 and VEGF signaling pathways crosstalk	Upregulated	0.004492
	2	VEGFA-VEGFR2 Signaling Pathway	Upregulated	0.01316
	3	Angiogenesis	Upregulated	0.01786
	4	Canonical and non-canonical Notch signaling	Upregulated	0.02007
	5	Epithelial-to-mesenchymal transition in cancer	Upregulated	0.1129
	6	Ras Signaling	Upregulated	0.1295
	7	Focal adhesion	Upregulated	0.1393
	8	Xenobiotic metabolism	Upregulated	0.14
	9	Focal adhesion-PI3K-Akt-mTOR-signaling pathway	Upregulated	0.2047
	10	PI3K-Akt Signaling Pathway	Upregulated	0.2269
**DOK Spheroids**	11	KRAS Signaling	Upregulated	0.00001215
	12	Pathways in cancer	Upregulated	0.0000309
	13	Neovascularisation processes	Upregulated	0.0001093
	14	Matrix metalloproteinases	Upregulated	0.0002256
	15	Robo4 and VEGF signaling pathways crosstalk	Upregulated	0.004492
	16	Coagulation	Upregulated	0.004679
	17	Benzo(a)pyrene metabolism	Upregulated	0.006731
	18	PI3K/AKT/mTOR signaling	Upregulated	0.07595
	19	Fatty acid metabolism	Upregulated	0.1122
	20	Apoptosis modulation and signaling	Downregulated	0.000001837

**Table 3 ijms-24-04096-t003:** Quantitative assessment of invasion in the chemoprevention assay performed on transformed HOK and DOK spheroids treated with pazopanib and lenvatinib using one-way ANOVA.

	Transformed HOK Spheroids		Transformed DOK Spheroids	
	Length (µm)	Area (µm^2^)	Length (µm)	Area (µm^2^)
**Control**	341.6 ± 40.9	50,718 ± 814.0	614.2 ± 39.4	125,235.7 ± 3158.1
**Pazopanib-treated**	153.8 ± 29.3 *	14,862.8 ± 2683.2 *	373.8 ± 49.3 *	71,685.5 ± 16,675.1 *
**Lenvatinib-treated**	131.1 ± 20.4 *	15,486.0 ± 4136.3 *	341.3 ± 20.0 *	49,848.6 ± 27,842.6 *

* *p*-value less than 0.05.

## Data Availability

The data presented in this study are available on request from the corresponding author.

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
