# Peer review of "A Novel Preclinical In Vitro 3D Model of Oral Carcinogenesis for Biomarker Discovery and Drug Testing"

_ijms, 2023, doi:10.3390/ijms24044096_

Round 1
Reviewer 1 Report (Previous Reviewer 2)
Standardization of the reference to journals names is still required (names in full or abreviated names?)
Citation 8: the number of the article is still missing.
Author Response
Thank you for your time in reviewing our work. References were checked an updated
Reviewer 2 Report (Previous Reviewer 1)
The authors appropriately addressed my previous comments. Manuscript is suitable for a publication in IJMS.
Author Response
Thank you for reviewing our work.
Reviewer 3 Report (New Reviewer)
Review on Shuryaprakash et al’s study entitled “A novel in vitro model of oral carcinogenesis identifies the role of vascular epithelial growth factor signalling in oral carcinogenesis”
1. The importance of the study’s subject can be emphasized more with epidemiological data in the introduction. Introducing prevalence of OSCC and smoking e.g. in Australia, or if data are not available than in the world would make your work more specific.
2. The methods in the materials and methods part sometimes are not described accurately. The source of HOK and DOK cells is missing from the description.
3. Please, describe how many plates have you used for this certain experiment stage. Bet, triplicate a day? Give a more followable description of the process.
4. Why imaging and morphometric analysis had been undertaken 1-8 days for HOK, while 2-6 days for DOK? Please, insert explanation to the text.
5. A few abbreviations are not in the correct way in the text. Where mentioned first, please supplement those abbreviations in brackets!
6. The relationship between IRS and the presence of VEGF was examined by histology and immunoreactivity assessments. Please state this at the beginning of this method’s description. it would be more understandable. Not all your readers are fully aware of the purpose of certain procedures.
7. IRS is a semi-quantitative index, practically. In case of these type of indices the calibration is advisable before evaluation, not by the end of it. In clinical periodontology this is an epidemiologically accepted method if examinations are done by multiple examiners.
8. Variables used in statistical evaluation are described in “Results”. It would be better to give a short description of the variables used in statistical evaluation in the “Statistical analysis” section though the stages of IRS is more detailed here than needed as it has been described in the previous paragraph. In this part it is enough to mention that it is a categorical variable that can take the values of 1,2,3 and 4.
9. Please, introduce the true p values to the text.
10. The use of spheroids is a good option in testing new drugs before starting clinical tests. With the help of these models it easier to pick the suitable applicants for clinical testing.

Author Response
We thank the reviewer for their time in reviewing our work.
- The importance of the study’s subject can be emphasized more with epidemiological data in the introduction. Introducing prevalence of OSCC and smoking e.g. in Australia, or if data are not available than in the world would make your work more specific.
Thanks for the suggestion. We added a couple of statements on oral cancer in the world and Australia.
- The methods in the materials and methods part sometimes are not described accurately. The source of HOK and DOK cells is missing from the description.
The method section has been revisited and updated.
- Please, describe how many plates have you used for this certain experiment stage. Bet, triplicate a day? Give a more followable description of the process.
All experiments were done in triplicate. The whole section has been updated.
- Why imaging and morphometric analysis had been undertaken 1-8 days for HOK, while 2-6 days for DOK? Please, insert explanation to the text.
Good point. A statement to explain that was added.
- A few abbreviations are not in the correct way in the text. Where mentioned first, please supplement those abbreviations in brackets!
All anomalies are corrected.
- The relationship between IRS and the presence of VEGF was examined by histology and immunoreactivity assessments. Please state this at the beginning of this method’s description. it would be more understandable. Not all your readers are fully aware of the purpose of certain procedures.
This has been revised.
- IRS is a semi-quantitative index, practically. In case of these type of indices the calibration is advisable before evaluation, not by the end of it. In clinical periodontology this is an epidemiologically accepted method if examinations are done by multiple examiners.
In fact, in our lab as standard operation procedure, all IHC staining is assessed independently by two examiners. A consensus was achieved between the two examiners.
- Variables used in statistical evaluation are described in “Results”. It would be better to give a short description of the variables used in statistical evaluation in the “Statistical analysis” section though the stages of IRS is more detailed here than needed as it has been described in the previous paragraph. In this part it is enough to mention that it is a categorical variable that can take the values of 1,2,3 and 4.
Thanks. Revised as you suggested.
- Please, introduce the true p values to the text.
Exact p values were added.
- The use of spheroids is a good option in testing new drugs before starting clinical tests. With the help of these models it easier to pick the suitable applicants for clinical testing.
Thanks for your statement. This has been incorporated to the title and discussion
This manuscript is a resubmission of an earlier submission. The following is a list of the peer review reports and author responses from that submission.
Round 1
Reviewer 1 Report
In this study, Suryaprakash et al were able to establish a 3D model of oral carcinogenesis by using normal and dysplastic spheroids and by treating them with the known carcinogen 4NQO. They also identified some potential pathways associated with progression to oral cancer. Their model would be suitable for testing a range of chemotherapeutic agents which may be useful as a preclinical model for OSCC. However, here some comments to be addressed by the researchers.
A. The title does not reflect the methodology of this study as the role of VEGF signaling in OSCC was not validated by any further studies.
B. Despite many of the pathways associated with tumor angiogenesis significantly upregulated. However, the authors did not validated any of these genes to support their final conclusion.
C. The authors did not attempt to explore the role of VEGF in the transition from normal oral mucosa, through differing degrees of dysplasia, to invasive carcinoma to make it clinically relevant.
D. A further elaboration is needed about how exactly this model will replace the use of animal models for both research and clinical purposes.
E. In their next generation sequencing (NGS), the authors used 4NQO, this carcinogen is well-known to replicates the synergistic effects of tobacco and alcohol associated OSCC. However, the aspects of alcohol/and or Tobacco-induced oral carcinogenesis and cell transformation were not sufficiently discussed in the discussion section as some molecular changes were clearly documented in the literature to be detected in clinically apparent oral carcinomas, such as the loss of E-cadherin, gains in vimentin, or the nuclear abundance of YAP.
F. The section of the “Conclusions” is quite poor, it should be reinforced with comments on future prospects regarding how their model is suitable for testing a range of chemotherapeutic agents I a clinical settings.
Minor point:
1. The figures are not clear enough and should be replaced by high-resolution figures.
Author Response
We thank the reviewer for the invaluable comments and essential feedback.
Since the reviewer raised the importance of validating the changes in the expression of VEGF in cases representing normal oral mucosa, different grades of epithelial dysplasia and oral squamous cell carcinoma, we underwent an IHC staining of VEGF in a series of FFPE sections. We revised methods, results and discussion after this change. Therefore, the study findings support the title and provide evidence of the role of VEGF.
The discussion and conclusion were revised to improve their clarity.
We also replaced the photographs with high-resolution images.
Reviewer 2 Report
Int J Mol Sci – Suryaprakash et al
The authors propose a 3D cell culture model of oral carcinogenesis using spheroids that were considered representative of tumor microenvironment, treating them with NQO and employing transcriptomic analysis to identify potential pathways of oral squamous cell carcinoma. Based on results involving identification of spheroid viability, 3D invasion properties, chemopreventive effects and the expression of genes associated with tobacco-induced oral squamous cell carcinoma, the authors concluded on the clinically relevance of their proposed model. The manuscript is original, concisely written, technically well done, and suitable for the journal.
Minor:
Figures 1-3 – substitute letters a, b, … for A, B,…to be consistent with the illustrations
To agree with the majority of the references’ format, the identification of the name of the journals in full should be mentioned for the citations 8, 19, 22, 23, 26, 39, 40, 41
Citation 8: the article number is missing
Citation 33: correct the 3rd line
Citation 45: Is it necessary to inform all co-authors?
Author Response
Many thanks for your supportive comments. We checked the manuscript and corrected all anomalies that we found, including the ones that you highlighted.